# The Psychosocial and Somatic Effects of Relocation from Remote Canadian First Nation Communities to Urban Centres on Indigenous Peoples with Chronic Kidney Disease (CKD)

**DOI:** 10.3390/ijerph18073838

**Published:** 2021-04-06

**Authors:** Denise Genereux, Lida Fan, Keith Brownlee

**Affiliations:** 1School of Social Work, Lakehead University, Thunder Bay, ON P7B 5E1, Canada; lfan@lakeheadu.ca; 2School of Social Work, Lakehead University, Orillia, ON L3V 0B9, Canada; kbrownle@lakeheadu.ca

**Keywords:** chronic kidney disease, renal failure, end-stage renal disease, dialysis, indigenous, Northern, Canadian, rural, remote, urban, relocation

## Abstract

Chronic kidney disease, also referred to as end-stage renal disease (ESRD), is a prevalent and chronic condition for which treatment is necessary as a means of survival once affected individuals reach the fifth and final stage of the disease. Dialysis is a form of maintenance treatment that aids with kidney functioning once a normal kidney is damaged. There are two main types of dialysis: hemodialysis (HD) and peritoneal dialysis (PD). Each form of treatment is discussed between the patient and nephrologist and is largely dependent upon the following factors: medical condition, ability to administer treatment, supports, geographical location, access to necessary equipment/supplies, personal wishes, etc. For Indigenous Peoples who reside on remote Canadian First Nation communities, relocation is often recommended due to geographical location and limited access to both health care professionals and necessary equipment/supplies (i.e., quality of water, access to electricity/plumbing, etc.). Consequently, the objective of this paper is to determine the psychosocial and somatic effects for Indigenous Peoples with ESRD if they have to relocate from remote First Nation communities to an urban centre. A review of the literature suggests that relocation to urban centres has negative implications that are worth noting: cultural isolation, alienation from family and friends, somatic issues, psychosocial issues, loss of independence and role adjustment. As a result of relocation, it is evident that the impact is profound in terms of an individuals’ mental, emotional, physical and spiritual well-being. Ensuring that adequate social support and education are available to patients and families would aid in alleviating stressors associated with managing chronic kidney disease.

## 1. Introduction

According to The Kidney Foundation of Canada [1], approximately 1 in 10 Canadians have kidney disease and millions more are at risk. In fact, the number of individuals living with kidney disease has increased 37% since 2007 and approximately 48,000 Canadians are receiving treatment for kidney failure [1]. Unfortunately, Indigenous Peoples, who compromise 4.3% of Canada’s population (over 1.4 million people), have a disproportionately high rate of kidney disease in comparison to non-Indigenous Peoples [2,3,4,5,6,7]. Statistics demonstrate that between the years 1980 and 2000, the number of Indigenous peoples receiving dialysis increased 8-fold [5,8]. Additionally, Indigenous Peoples are four times more likely to experience end-stage renal disease (ESRD) than non-Indigenous People [5,6,9]. According to Sood and colleagues, the Indigenous Peoples of Canada experience “*a high burden of illness, low socio-economic status and geographic isolation*” ([10], p. 1433). Consequently, this contributes to an increased prevalence of diabetes mellitus, obesity and hypertension which are some of the leading risk factors to kidney disease and renal failure. This combination of medical and societal factors, in conjunction with delayed access to a nephrologist, contribute to the increasing prevalence of ESRD [5]. Unfortunately, it is evident that individuals who reside in remote locations who are in need of dialysis treatment often have limited choice and frequently have to leave their communities and relocate to urban centres [9]. Consequently, the objective of this paper is to determine the psychosocial and somatic effects of relocation from remote Canadian First Nation communities to urban centres on Indigenous Peoples with chronic kidney disease (CKD).

## 2. Chronic Kidney Disease (CKD)

Chronic kidney disease (CKD), also referred to as chronic kidney failure (CKF), occurs when the functioning of a kidney progressively deteriorates. Kidneys function to filter wastes and excess fluid from the blood, which are later excreted in the urine. However, as CKD progresses and kidney functioning declines, harmful levels of fluid, electrolytes and wastes can accumulate in our bodies which can be detrimental to our health [11]. A diagnosis of CKD puts individuals at an elevated risk of cardiovascular diseases which include increased chances of cardiac arrest or stroke.

### 2.1. Stages of Chronic Kidney Disease (CKD)

CKD consists of five stages, ranging from mild to severe loss of kidney function. This is determined based on an individual’s glomerular filtration rate (GFR) which is a measurement (in milliliters per minute) that determines the rate at which kidneys filter wastes and extra fluid from the blood [12]. Additional indicators of CKD (which must be present for approximately three months) include the presence of albuminuria in urine, urine sediment abnormalities, electrolyte imbalances, histologic abnormalities, structural abnormalities, and/or history of kidney transplant [13]. When kidney functioning deteriorates to approximately 30% of normal, individuals with CKD often experience symptoms of uremia which includes “weakness, nausea, loss of appetite and a bitter taste in the mouth” [14] (p. 3). However, when kidney functioning declines to 10–15% of normal, this indicates that the individual has reached the fifth and final stage of CKD, also referred to as End Stage Renal Disease (ESRD) or kidney failure. At this stage, individuals require treatment for survival [14]. For further information regarding the stages of CKD and the diagnostic criteria, please refer to the KDIGO 2012 Clinical Practice Guideline for the Evaluation and Management of Chronic Kidney Disease ([13], p. 5).

### 2.2. Forms of Treatment

End-stage renal disease (ESRD) is an incurable “*progressive and debilitating chronic illness*” that requires treatment ([15], p. 1312). There are two forms of treatment available: dialysis or kidney transplant. For the purposes of this paper, dialysis treatment will be the primary focus.

Dialysis is a form of treatment that replaces some of the functioning of a normal kidney when it becomes damaged [16]. Dialysis prevents toxic build-up in our bodies by filtering waste, salt and extra water and by keeping a safe level of certain chemicals in our blood (i.e., potassium, sodium and bicarbonate) [17]. As previously mentioned, dialysis is typically required when the individual experiences kidney failure (ESRD) or when 10–15% of their kidney functioning is remaining. At this point, the individual would require dialysis for the rest of their life as maintenance, or receive a kidney transplant if deemed eligible [14]. Dialysis treatment does not have a reverse effect on kidney function. Unfortunately, there is no cure [18,19]. Individuals typically receive dialysis in hospital settings, on a dialysis unit that is not affiliated with a hospital, or in their home. The nephrologist (a physician who specializes in kidney functioning) and patient collaboratively determine the best location to receive treatment based on the individual’s medical condition and personal wishes [16]. However, if the individual is incapable of performing dialysis independently or lacks support from family or friends, hospital or dialysis units are strongly recommended due to the complexity of the dialysis routine [17]. Lastly, unless a donor kidney is readily available for an eligible patient, that individual would require a form of dialysis as they await surgery [9,18]. This option does not appear to be favourable to some cultures, including Indigenous Peoples, due to perceptions and beliefs regarding organ donation [20].

Hemodialysis (HD) and peritoneal dialysis (PD) are the two main types of dialysis (Figure 1). According to Murphy and colleagues, these forms of treatment are “*very different in terms of technique and physiology*” ([21], p. 2557). Each method of dialysis has differing risks and benefits [10] and there is very little evidence that indicates whether either method is clinically superior [22]. Both forms of treatment require close and continuous monitoring by a nephrologist to ensure effective treatment and to address any complications that may arise [23]. Again, the type of dialysis a patient receives is dependent upon discussion between the patient and nephrologist. Additional factors that may influence treatment modality include the patient’s ability to adequately perform treatment, additional medical comorbidities, or geographic location [21]. Evidently, the process of determining the best treatment modality for patients can be complex due to each individual’s unique needs and circumstances.

#### 2.2.1. Hemodialysis (HD)

HD is a form of treatment that cleans blood outside of the body. In this process, a machine and filter are used to remove waste products and excess water from the blood [16]. Hemodialysis is reportedly the most common treatment for patients with ESRD [4,14]. In fact, approximately 85% of patients with ESRD in Northeastern Ontario use this form of treatment [4]. HD patients typically receive treatment in a hospital or community clinic three times per week (Monday, Wednesday, Friday, or Tuesday, Thursday, Saturday) for approximately four hours at a time, although this can range from three to five hours [14,15,16,17,18,19].

#### 2.2.2. Peritoneal Dialysis (PD)

PD cleans the blood internally utilizing a fluid (dialysate) that is inserted into the abdomen via a catheter to discard waste products and excess fluid from the body [16]. In Canada, approximately 18% of all dialysis patients receive PD as treatment [10]. PD enables patients to be treated in their own homes, which promotes independence. However, PD patients are also periodically seen in an ambulatory clinic setting for review [10]. Two forms of PD are available: Continuous Ambulatory Peritoneal Dialysis (CAPD) and Automated Peritoneal Dialysis (APD) [17].

CAPD requires patients to hook the dialysate to their catheter manually to drain the fluid once the exchange is complete. While this treatment allows patients to complete normal activities, CAPD does require three to five thirty to forty minute exchanges every twenty four hours [17].

In contrast, APD is completed via a machine (or cycler) that delivers and drains the dialysate automatically. Administration occurs during the night while the patient is sleeping. Evidently, the only difference each form of PD treatment is the number of treatments and the process in which the treatment is completed [17].

## 3. Dialysis and Northern First Nation Communities

Based on the descriptions of both types of dialysis (PD and HD), it becomes apparent that the treatment options available are complex and do not come without challenges. For instance, dialysis requires dedication and commitment from patients who must comply with treatment protocols and procedures. Both forms of dialysis are time consuming and require adherence to strict dietary and fluid restrictions in conjunction with medication compliance [19]. This profound modification of lifestyle contributes to the numerous losses that one can experience (i.e., loss of health, independence, identity, etc.). Although dialysis in itself is associated with several losses, this is further complicated for individuals who reside on remote Canadian First Nation communities.

### 3.1. Quality of Living

According to Wilson and colleagues [6], health care is more than addressing symptoms alone, especially considering the number of resources and supplies dialysis treatments require. Other elements that must be considered include an individual’s quality of living, their housing conditions and quality of water. For Indigenous peoples residing on Canadian First Nation communities, they are more likely to have “*poor housing with 26% living in crowded homes and 44% of homes [requiring] major repairs*” ([24], p. 654). In fact, the literature indicates that living conditions on First Nation communities are amongst the poorest in Canada [25]. For instance, the water quality on many First Nation communities does not meet Canadian drinking water standards and some homes do not have running water [26]. This is further complicated with a lack of available plumbing or electricity [24].

### 3.2. Geographical Location

Another complicating factor of living on First Nation communities is the fact that many of these remote locations are only accessible via air for most of the year. Winter roads that are created over frozen lakes and marshes are only accessible for several months per year, weather conditions depending [24]. Since air freight is a costly method of delivering supplies to remote First Nation communities and is only used in emergency situations, supplies are typically delivered via winter roads once per year [24]. Evidently, the limited options available in terms of accessing supplies indicates that resources for dialysis are scarce on First Nation communities.

Furthermore, the inaccessibility and geographical location of many First Nation communities further complicates and limits access to ambulatory, acute and specialized health care [3,27]. Health care on remote Canadian First Nation communities is typically limited to a nursing station and visits by itinerant physicians [24]. Although there are several health-related benefits to providing medical services close to home, other elements such as expense, training of health care staff and specialist availability make receiving dialysis much more difficult in isolated communities [6]. Consequently, recipients of dialysis treatment who reside in remote locations are often required to travel to urban centres to access medical specialists and services [28]. In some instances, absence from communities is only temporary, ranging from several hours to days; however, at other times, absence is much longer.

### 3.3. Medical Relocation

According to Lavoie and colleagues, medical relocation refers to “*situations where First Nation peoples relocate to an urban centre for an extended period of time to access care*” ([28], p. 296). Relocation to urban centres is often necessary for medical care pertaining to end-of-life needs or for dialysis if this cannot be adequately performed in an individual’s home community [6,28]. A significant statistic from the Canadian Institute for Health Information indicates that “*one in five Aboriginal patients with ESRD was required to travel more than 250 km to a health care facility to receive treatment, compared with less than 5% of non-Aboriginal patients*” ([20], p. 5). Since geographic location may increase difficulty in both providing and receiving high quality treatment, this may impact an individual’s overall health outcome [22]. For instance, limited access to health care on First Nation communities increases the chances of dialysis patients reaching ESRD. As a result, these patients are “*more likely to die or be hospitalized when compared to individuals who live closer to specialized care*” ([3], p. 5). A study by Shah et al. confirms this when they indicate that Indigenous Peoples living in rural areas “*have higher hospitalization rates compared with their urban counterparts*” ([29], p. 798).

### 3.4. Selection of Dialysis Treatment

Selecting the most appropriate form of dialysis treatment for patients evidently becomes increasingly difficult when restrictions to geographical locations are apparent. Specifically, Zacharias and colleagues [24] make note of the challenges with establishing home HD in remote communities (i.e., poor or inadequate housing, unreliable water supply, limited medical staff, and poor access to the community, etc.). Due to the many resources this form of treatment requires (i.e., trained staff, dialysis machines, etc.) and the demanding frequency of treatments (three times per week for several hours at a time), it becomes essential to live in close proximity to urban centres [9,10].

Since PD is considered a “home-based therapy,” this form of dialysis treatment is said to be ideal for individuals who reside in rural or remote locations to avoid relocation [5,10,22]. According to Mathew et al. [5], roughly 64% to 75% of patients diagnosed with ESKD qualify for PD. Additionally, individuals who resided between 50–300 km away from urban centres in comparison to individuals living within 50 km were more likely to initiate PD treatment [22]. Although PD permits individuals to independently administer treatment while completing everyday activities or while sleeping, this option is not always suitable for individuals living on First Nation communities.

Again, PD requires access to adequate water, electricity and supplies; however, these necessities are not always available in remote communities [9]. PD treatment also requires education, adequate medical support, and dedication from both the patient and their support network to comply with treatment recommendations [9]. In contrast, physical debilitation, lack of motivation and/or commitment, and lack of social and/or family support are additional factors that hinder patients from this treatment modality [5,6,7,8,9,10,11,12,13,14,15,16,17,18,19]. According to a study conducted by Tonelli and colleagues [8], U.S. nephrologists identified patient compliance as the most essential factor when recommending a dialysis modality. Unfortunately, the same study found that “ethnic minorities may be considered less likely to adhere to medical advice,” likely due to cultural barriers [8] (p. 487). Furthermore, due to the complexity of the dialysis routine, extra assistance from family members is paramount. A lack of understanding or support from families or friends can be detrimental as it can lead to complications such as technique failure or peritonitis [8,10]. As a result, HD may be the recommended treatment modality along with relocation to an urban centre to ensure adequate care is being provided [10].

## 4. Methods

Several databases were explored as part of the search strategy: CINAHL, Google Scholar, Medline, Proquest, PsychInfo, Pubmed, and Social Services Abstracts. Search terms comprised of “Chronic Kidney Disease” OR “CKD” OR “Renal Failure” OR “End Stage Renal Disease” OR “ESRD” OR “Renal” OR “Dialysis” AND “Aboriginal” OR “Indigenous” OR “Native American” OR “First Nation” AND “Northern” OR “Canadian” AND “Rural” OR “Urban” AND “Reserve” AND “Relocation” AND “Mental Health” OR “Effect” OR “Mood.” Limiter included the term “Australian Aboriginal” as the focus of this study was on Indigenous populations on remove First Nation communities. Lastly, the articles selected were based on their contents’ relevance to the objectives of this paper.

## 5. Literature Review

Upon review of the literature, scholars were able to determine that relocation did have a profound effect on Indigenous Peoples with CKD who relocated from rural to urban centres. Several common themes emerged throughout the literature: cultural isolation [4,6,9,24,27,30], alienation from family and friends [4,6,9], somatic issues [4,9], psychosocial issues [4,6,7,9,15,18,24,31,32], and lastly, loss of independence and role adjustment [4,9,18,19,33].

### 5.1. Cultural Isolation

Relocation from rural home communities to unfamiliar, urban centres can be a foreign and frightening experience for some individuals. This is especially true when an individual does not feel that their culture is accepted or included in the new community. Literature indicates that relocating individuals to urban centres for dialysis treatment can be culturally stressful to patients [4,6,9,24]. More specifically, differences between the individual’s home community and urban centres can be so profound that they can experience culture shock. A study conducted by Kolewaski and colleagues found that dialysis patients would describe themselves being “‘*uprooted’ and ‘pulled’ from their homes and cultural environments*” ([4], p. 122). Participants found adapting to urban centres is an overwhelming experience and further described this transition as “*moving from ‘heaven to hell’*” ([4], p. 122). Evidently, cultural isolation had a profound impact on the participants in this study as they were removed from their social supports and familiar contexts.

Wilson and colleagues [6] further argue that urban health care centres that are unfamiliar and lack cultural sensitivity hinders compliance and treatment outcome. Shah and Farkas support this when they acknowledge that health concerns are further complicated with “*stress of adaptation to urban living*,” “*unfamiliarity with urban health care systems*,” and communication problems ([30], p. 860). For example, differences in culture and values and beliefs pertaining to health influence adherence to medical interventions and treatments [4].

Similarly, communication between patients and staff is crucial in the health care setting [27]. Scholars found that patients lacked information regarding their health care which thereby compromised their understanding and intensified feelings of culture shock [4,6]. Evidently, language barriers are a significant hindrance to accessing adequate health care [27]. Despite language barriers hindering communication, patients of the study were expected to be responsible for being informed regarding their care [4]. However, Maronne acknowledges that even if both the patient and the provider “*speak the same language, the cultural values and experiences of the patient influence how they communicate their symptoms and how they perceive feedback about their health status from the provider*” ([27], p. 193). Consequently, cultural sensitivity is a significant component in ensuring patient care needs are being met. For this reason, it is imperative that language interpreters and educational resources are available to support patients in understanding treatment interventions and communicating with health care providers [20].

Relocation and dialysis treatment also impacted Indigenous People’s culture through recreational pursuits. Traditionally, Indigenous Peoples enjoy “*outdoor, land-based activities such as hunting, fishing and camping*” ([9], p. 22). Seasonal hunts and activities provided excellent opportunities for social gatherings [4]. However, upon relocation to urban centres, many patients do not feel that they could enjoy these same activities due to the urban location, their physical health, or other related factors [9].

### 5.2. Alienation from Family and Friends

One of the more apparent negative effects of relocation is alienation from family and friends. According to a study conducted by Wilson and colleagues [6], many patients shared feelings of homesickness and struggled with separation from family and friends. Kolewaski and colleagues [4] further support this when they identified that family separation was one of the most difficult things dialysis patients had to cope with as a result of relocation. As a result of this separation, it was found that the patients’ emotional and social support needs were not being met [4]. Dialysis patients would worry about their children being able to cope on their own and often felt disconnected from their family members and friends [9]. It was also found that the loneliness and isolation imposed by relocation further influenced dialysis patients’ quality of life [4,6].

Unfortunately, many dialysis patients who relocate to urban centres miss many social and cultural events (i.e., funerals, birthday celebrations, graduations, hunts, etc.) which once made them feel connected to the community [4]. Some dialysis patients are able to cope with the distance by having return visits home if permitted by their health team. Although return visits home were beneficial in terms of patients’ mental and social well-being, their physical health often suffered as a result of noncompliance with the dialysis regime [4]. At the same time, heath care providers would recognize that patients’ homesickness was at “crisis level” [4]. As a result of having the opportunity to reconnect with family and friends, Salvalaggio and colleagues noted that “*the hardships of dialysis were minimized and quality of life improved*” ([9], p. 22). However, returning to urban centres for treatment after a short visit was noted to cause “*significant emotional distress for the patients*” ([4], p. 128).

Moreover, although having friends and family visit dialysis patients removed the health risk of patients flying home, financial hardship often ensued [4]. Salvalaggio and colleagues also acknowledged that financial burden is a reality for dialysis patients who wish to temporarily return home by indicating that “*a lot of people can’t go home because they can’t afford it*” ([9], p. 22). Coordinating return visits home or to urban centres is further complicated by school or work commitments, travel distances, and weather. Ultimately, Kolewaski and colleagues [4] found that individuals who are surrounded by a strong support network had more favourable health outcomes in comparison to those with a limited support network.

### 5.3. Somatic Issues

The restrictive nature of dialysis treatment in conjunction with the medical diagnosis itself can have a significant physical impact on patients. For example, symptoms of weakness and fatigue are a typical consequence of treatment [9]. Moreover, the experience of pain during the course of illness and treatment negatively impacts the individual’s ability to complete formerly easy tasks (i.e., activities of daily living) and hinders enjoyment in life [9]. Although the restrictive nature of dietary options and fluid consumption is a difficult adjustment for non-Indigenous individuals, this becomes more challenging for Indigenous Peoples as they are advised to avoid traditional foods such as wild game (i.e., moose, deer, beaver, etc.) [4]. Noncompliance of dietary suggestions may lead to nausea and feelings of general discomfort and unwellness. Other physical symptoms include shortness of breath, body swelling, and abnormal smells or tastes [9]. Individuals with CKD or ESRD often struggle with eating meals due to physical symptoms which may cause a loss of appetite. Issues concerning mental health and alienation from family and friends contribute to poor appetite and somatic symptoms as health is a multidimensional construct. Kolewaski and colleagues support this when they acknowledge that dialysis treatment “*directly affected the perceptions of health, family life, daily activities, and social roles due to the fatigue and […] treatment effects they experienced*” ([4], pp. 116–117).

### 5.4. Psychosocial Issues

According to Peiris and colleagues [32], psychosocial stress is common amongst vulnerable populations as it hinders access to health care and is associated with adverse health outcomes for Indigenous people. Indigenous Peoples and their families who reside on remote First Nation communities are confronted with “*major psychosocial disruptions of relocation*” when their loved ones must move to urban centres for dialysis treatment ([7], p. 756). Relocating for medical treatment “*disrupts social support patterns*” ([6], p. 1934) and is “*culturally stressful to patients*” ([24], p. 653). In turn, literature has found that this had a profound impact on patients’ mental health [4,9,15,18,31]. For instance, Haverkamp and colleagues [31] found that symptoms of depression and anxiety were common amongst chronic dialysis patients. Consequently, the participants in their study were more prone to “*an impaired quality of life, increased hospitalization and increased mortality*” ([31], p. 26). Issues pertaining to marital relationships was also identified as a significant psychosocial impact [4]. White and Grenyer noted that dialysis patients and their families were “*overwhelmed by the impact of dialysis on their lives*” ([15], p. 1312). Feelings of anger, depression, hopelessness, sadness, resentment, guilt and loss were prevalent amongst dialysis patients and their partners [15]. Additionally, Haverkamp and colleagues [31] acknowledged that acculturation has a significant influence on patient’s mental health. Grief and loss are common feelings experienced by dialysis patients. This is a result of the numerous changes associated with the treatment regimen in conjunction with relocation and readjustment to the patient’s “new normal.” As a coping mechanism, patients may respond to this with denial or noncompliance with treatment, thereby affecting their overall health outcome [9].

### 5.5. Loss of Independence and Role Adjustment

Recipients of dialysis rely on this form of treatment for their physical health and overall well-being. Without proper medical interventions, their health would deteriorate significantly. As previously mentioned, dialysis patients are advised to comply with a strict treatment regime that is accompanied with a set of rules in terms of diet, fluid intake, blood sugar levels, medications, and exercise. In other words, patients are told “*’what to do’ and ‘what not to do’*” by their health care team ([33], pp. 108–109). Salvalaggio and colleagues [9] support this when they recognize that dialysis patients often struggle with loss of autonomy regarding their medical care. According to Curtin and colleagues, “*maintaining any kind of normal lifestyle in the face of this commitment is difficult and demands the profound rearranging of [daily] activities*” ([19], p. 621). Consequently, loss of independence and change in both personal and professional roles that once provided a sense of pride and accomplishment are common amongst dialysis patients. Two prime examples of this are in terms of caregiving or employment. Specifically, patients have noted role reversal in terms of caregiving (i.e., children are now the caregivers) and employment status may be lost due to relocation or physical demands [4,9]. Moreover, relocation for dialysis treatment not only created physical distance between patients and family and friends; it also impacted social roles and socialization [9]. Consequently, patients are forced to “*recast and adapt their accustomed ways of thinking about themselves and their habitual ways of acting and interacting in the world*” ([19], p. 610). On the other hand, some dialysis patients may not physically appear ill. As a result, Courts and Boyette [18] found that patients were often pressured by family and staff to continue living a normal life. The additional stressor of balancing expectations versus reality can be overwhelming for dialysis patients as they attempt to adjust to a “new normal” (Figure 2).

## 6. Discussion

Chronic illnesses such as ESRD demand lifestyle changes on both the behavioural and emotional sphere, thereby challenging both the patient and their families’ coping ability [18]. Evidently, both the disease and dialysis treatment have a significant impact on many aspects of an individual’s life [9]. This is further complicated by geographical location in reference to treatment centres in urban locations. According to Maronne [27], living in a rural location is a barrier to adequate health care regardless of individual race. Bello and colleagues further support this when they acknowledge that individuals with CKD who reside in remote locations were “*less likely to receive recommended quality care” and were “more likely to experience adverse health outcomes*” in comparison to those living closer to a nephrologist ([34], p. 3849). In comparison to non-Indigenous individuals, Indigenous Peoples who reside in rural Canadian settings were found to have a significantly higher prevalence of end-stage kidney disease (ESKD) [5].

Moreover, due to the high concentration of Canadian nephrologists in major urban locations, Tonelli and colleagues [22] highlight that the selection of treatment modality is dependent upon geographical location among other factors such as capacity to administer treatment, social supports, physical health, etc. The rather abrupt expectation to relocate as a result of one’s medical diagnosis and environmental factors can disrupt social support patterns and influence overall quality of life [6,14,18,21].

According to White and Grenyer, the biopsychosocial model upholds the belief that “*every human is a unique and complex interconnection between the physical, psychological and social aspects of their daily life*” ([15], p. 1313). In other words, becoming ill and receiving treatment is a multidimensional experience that encompasses all aspects of self (mental, physical, spiritual, and emotional). On this level, it is apparent that a diagnosis of CKD or ESRD can impact all four of these areas due to the many losses one can endure in conjunction with relocation (i.e., culture, relationships with family and friends, life roles, physical impairment, etc.) [15].

As a result of this, individuals with CKD or ESRD are compelled to adapt to their “new normal.” Gregory et al. [35] note that adjustment to the “new normal” undergoes continuous transformation as the individual attempts to cope with the illness and treatment trajectory. White and McDonnell [36] recognize that sudden lifestyle alterations can impair an individual’s sense of independence along with their core values and beliefs. Similarly, Kolewaski and colleagues [4] found that the complexities associated with relocating were described as insurmountable by patients of their study. This further alludes to the drastic impact that illness and relocation can have on individuals’ lives.

In a study conducted by Kolewaski and colleagues [4], the scholars found that a patient compared their experience of relocation and dialysis treatment to the residential school system. Specifically, the patient felt that the “*isolation from social supports, removal from community and culture adjustment to both urban and medical context, routine of strict [dialysis] guidelines, and feeling that there was no alternative for survival*” was parallel to the experience of those in residential schools ([4], pp. 112–113). Salvalaggio and colleagues [9] validate this when they indicate that dialysis patients from remote First Nation communities are being removed from their homes and are expected to adjust accordingly. Similarly, White and McDonnell [36] highlight the fact that dialysis patients have to endure significant changes in their lives to comply with a strict treatment regime that does not offer a cure. It becomes apparent that the life-long commitment and numerous losses that individuals on dialysis may experience can be overwhelming for those involved.

Due to the drastic changes that dialysis and relocation involve, it is crucial that both patients and families are properly educated regarding the patient’s medical condition and the treatment process. According to Kolewaski and colleagues [4], a lack of knowledge was a significant barrier to the acclimatization process for Indigenous patients receiving dialysis. For example, the information and educational materials provided to patients were only available in the English language. Ensuring effective communication is held between the patient and their health care team would alleviate feelings of stress and anxiety that often accompany health-related concerns. Also, having a better conceptualization of their medical condition would allow the patient to more easily adjust to their new lifestyle. Likewise, incorporating cultural values of Indigenous peoples into their care would be beneficial as Indigenous culture and values may conflict with Western medicine and health care delivery [27]. This would further assist with the acclimatization process. Lastly, ensuring patients are supported by a multidisciplinary team would provide a holistic approach to care. This would ensure all of the individual’s aspects of self (i.e., mental, physical, emotional, spiritual) are being addressed (Figure 3).

## 7. Future Implications

Since the objective of this paper was to examine the effects of relocation on Indigenous peoples with CKD, it would be pertinent to generalize the study amongst non-Indigenous peoples who reside in rural locations as well [6]. Similarly, Harasemiw and colleagues suggest that different geographic locations should be examined with reference to the “*disparities in diabetes and CKD that are faced by Indigenous Canadians*” ([3], p. 6). Moreover, examining the health needs of smaller communities through the use of qualitative studies can assist in terms of future health planning [6]. Prospective studies should address the barriers to implementing and maintaining PD treatment for individuals who reside in rural locations as studies regarding this are limited [5,8]. It would also be of interest to explore how colonization has impacted the lives of Indigenous groups on a sociological and biological level to determine how this has influenced the health status of Indigenous Peoples both in the past and present [27]. Although this is very complex, it would be interesting to note whether this has any influence in terms of access to health care amongst Indigenous versus non-Indigenous populations [27]. For instance, is fear a barrier to accessing care at nursing stations due to the potential transfer to urban centres or permanent relocation? Lastly, based on the findings of this paper, it would be of note to examine the role of a multidisciplinary team with respect to Indigenous patients with CKD as well as Indigenous spirituality and culture.

## 8. Conclusions

The nature of chronic kidney disease and its associated treatments are complex and require patients to comply with treatment in order to preserve their overall health status and prevent further decline. Unfortunately, the rigorous nature of the disease and treatment forces patients to undergo numerous changes in their routines while attempting to cope with the disease. Evidently, illness and treatment have a profound impact on all aspects of self, including mental, physical, spiritual and emotional. Although diabetes is associated with several losses, it is further complicated for individuals who originate from remote First Nation communities as they may experience cultural isolation, alienation from family and friends, somatic and psychosocial issues along with loss of independence and role adjustment. Ensuring that adequate social support and education are available to patients and families would aid in alleviating stressors associated with chronic kidney disease. Moreover, utilizing a culturally sensitive approach and connecting patients to appropriate supports (such as a multidisciplinary team) may encompass a holistic approach to care to ensure all aspects of self (mental, physical, spiritual, and emotional) are well supported throughout the course of treatment.

## Figures and Tables

**Figure 1 ijerph-18-03838-f001:**
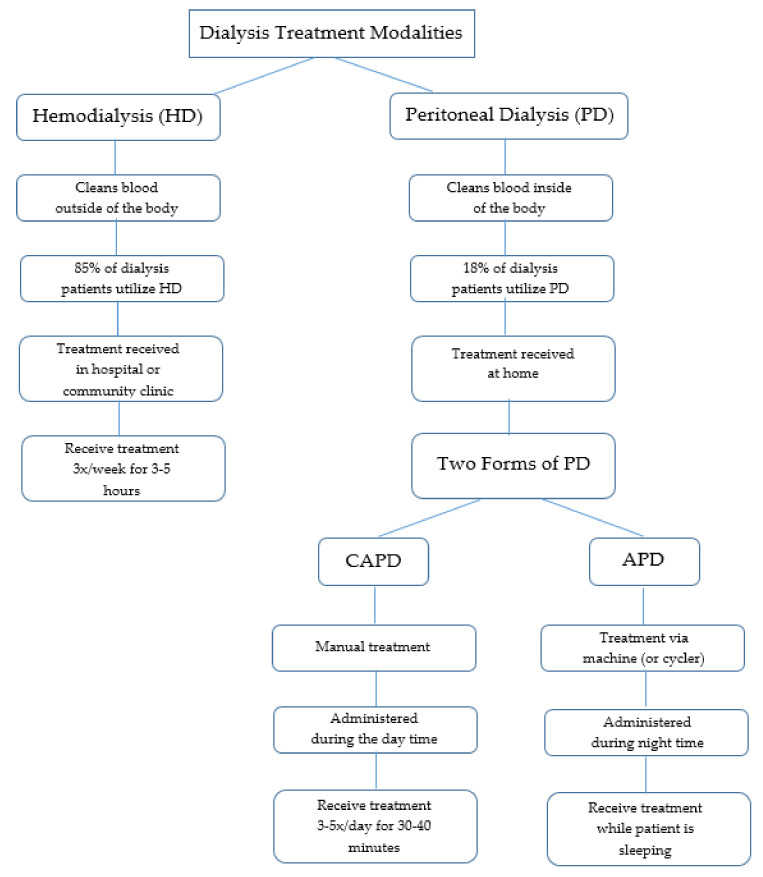
Overview of dialysis treatment modalities.

**Figure 2 ijerph-18-03838-f002:**
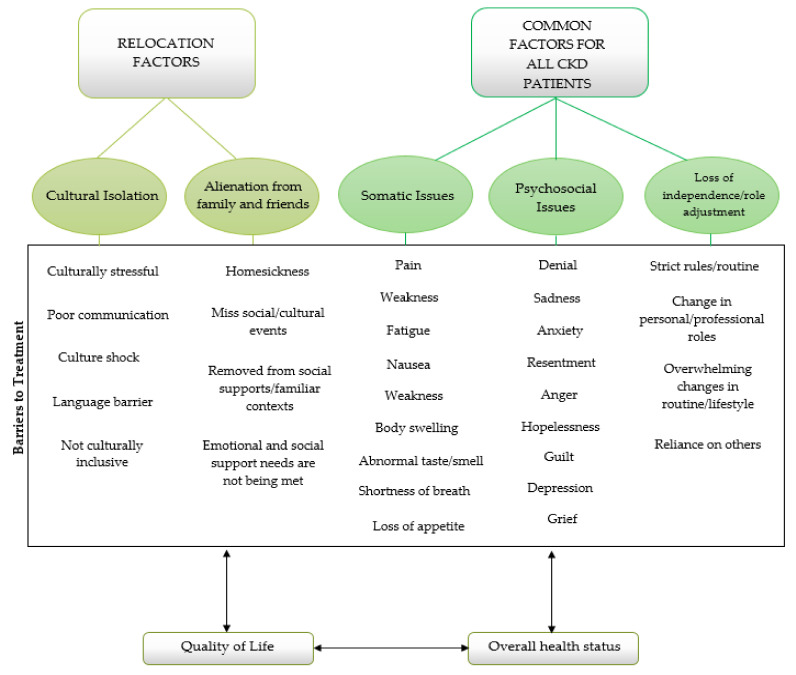
Interconnectedness of the effects of relocation and health outcomes.

**Figure 3 ijerph-18-03838-f003:**
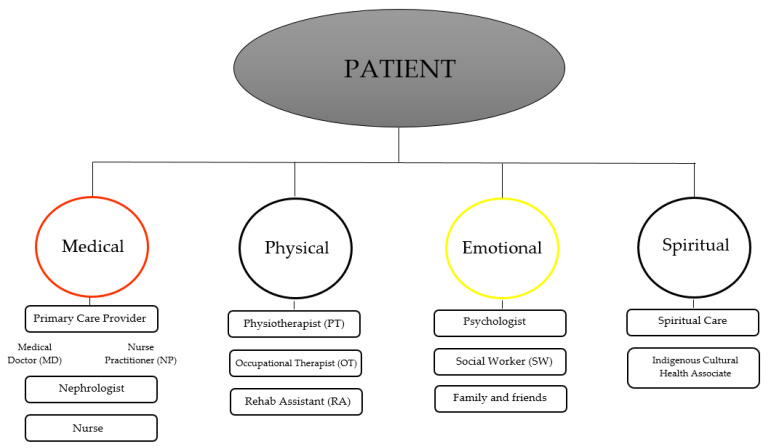
Illustrates the significance of a multidisciplinary team as it promotes a holistic approach to patient care.

## Data Availability

Not applicable.

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
