# Peer review of "The Psychosocial and Somatic Effects of Relocation from Remote Canadian First Nation Communities to Urban Centres on Indigenous Peoples with Chronic Kidney Disease (CKD)"

_ijerph, 2021, doi:10.3390/ijerph18073838_

Round 1
Reviewer 1 Report
Brief summary
The research illustrates the indirect effects of psychological and social nature of relocation on peoples with Chronic Kidney Disease (CKD), with a specific focus on Indigenous poeple living in Notherns Canadian Reserves. Indeed, relocation had profound effects on Indigenous peoples with CKD, and several common themes emerged throughout the literature: cultural isolation, alienation from family and friends, somatic issues, psychosocial issues, and loss of independence and role adjustment.
The reason is that becoming ill and receiving treatment is a multidimensional experience that encompasses all aspects of self
(mental, physical, spiritual, and emotional). As a result, individuals with CKD or ESRD are compelled to adapt to their “new normal”, undergoing continuous transformations as they attempt to cope with the illness and treatment trajectory. Although dialysis in itself is associated with several losses, this is further complicated for individuals who reside on Northern Canadian reserves for numerous reasons pertaining to the socioeconomic and cultural dimension of Indigenous life that are well-represented in the manuscript.
Lying on the results on the literature overview, the Authors suggest that it is crucial that both patients and families are properly educated regarding the patient’s medical condition and the treatment process. Furthermore, patients must be supported by a multidisciplinary team to ensure that all of the individual’s aspects of self (i.e., mental, physical, emotional, spiritual) are being addressed. Finally, the analysis of the role of a multidisciplinary team with respect to Indigenous patients with CKD as well as Indigenous spirituality and culture may deserve further research.
Broad comments
The manuscript addresses a crucial issue of socioeconomic nature, that is the additional indirect effect of CKD on Canadian Indigenous patients. As the topics are investigated exclusively on their qualitative dimension (i.e., the manuscript illustrates the results of an unstructured literature overview), Authors are kindly invited to summarize the main issues collected in three schemes or tables. The first scheme/table may illustrate the alternative forms of treatment of CKD (as an example, it may be a decision tree associating to each decision the most appropriate treatment) and may be placed before section 1.4. The second scheme/table may illustrate the indirect effects of relocation discussed in section 1.4 (cultural isolation, alienation, somatic issues...) and their main characteristics, and may be placed and the end of the section. Finally, the third scheme/table may summarize the outcome of the discussion and may be placed at the end of the section.
Specific comments
Authors are kindly invited to rethink the structure of the paper. Specifically, they may separate the introduction from 1.1. CKD and subsequent paragraphs; section 2 ("Objective") may be included in the introduction, while "Methods" may be placed before 1.1 CKD. The final structure: 1. introduction (and objective); 2. Method; 3. CKD and subsequent paragraphs; 4. Literature overview...
Author Response
March 30, 2021
Dear editor,
Thank you for giving us the opportunity to submit a revised version of our manuscript titled "The Effect of Relocation from Northern Canadian Reserves to Urban Centres on Indigenous Peoples with Chronic Kidney Disease (CKD)” to the International Journal of Environmental Research and Public Health.
Re: Comments for Manuscript ID ijerph-1162899 entitled "The Effect of Relocation from Northern Canadian Reserves to Urban Centres on Indigenous Peoples with Chronic Kidney Disease (CKD)."
Reviewer(s)' Comments to Author:
Reviewer: 1
Comments to the Author:
Brief summary
The research illustrates the indirect effects of psychological and social nature of relocation on peoples with Chronic Kidney Disease (CKD), with a specific focus on Indigenous people living in Northern Canadian Reserves. Indeed, relocation had profound effects on Indigenous peoples with CKD, and several common themes emerged throughout the literature: cultural isolation, alienation from family and friends, somatic issues, psychosocial issues, and loss of independence and role adjustment.
The reason is that becoming ill and receiving treatment is a multidimensional experience that encompasses all aspects of self (mental, physical, spiritual, and emotional). As a result, individuals with CKD or ESRD are compelled to adapt to their “new normal”, undergoing continuous transformations as they attempt to cope with the illness and treatment trajectory. Although dialysis in itself is associated with several losses, this is further complicated for individuals who reside on Northern Canadian reserves for numerous reasons pertaining to the socioeconomic and cultural dimension of Indigenous life that are well-represented in the manuscript.
Lying on the results on the literature overview, the Authors suggest that it is crucial that both patients and families are properly educated regarding the patient’s medical condition and the treatment process. Furthermore, patients must be supported by a multidisciplinary team to ensure that all of the individual’s aspects of self (i.e., mental, physical, emotional, spiritual) are being addressed. Finally, the analysis of the role of a multidisciplinary team with respect to Indigenous patients with CKD as well as Indigenous spirituality and culture may deserve further research.
Broad comments
The manuscript addresses a crucial issue of socioeconomic nature that is the additional indirect effect of CKD on Canadian Indigenous patients. As the topics are investigated exclusively on their qualitative dimension (i.e., the manuscript illustrates the results of an unstructured literature overview), Authors are kindly invited to summarize the main issues collected in three schemes or tables. The first scheme/table may illustrate the alternative forms of treatment of CKD (as an example, it may be a decision tree associating to each decision the most appropriate treatment) and may be placed before section 1.4. The second scheme/table may illustrate the indirect effects of relocation discussed in section 1.4 (cultural isolation, alienation, somatic issues...) and their main characteristics, and may be placed and the end of the section. Finally, the third scheme/table may summarize the outcome of the discussion and may be placed at the end of the section.
Specific comments
Authors are kindly invited to rethink the structure of the paper. Specifically, they may separate the introduction from 1.1. CKD and subsequent paragraphs; section 2 ("Objective") may be included in the introduction, while "Methods" may be placed before 1.1 CKD. The final structure: 1. introduction (and objective); 2. Method; 3. CKD and subsequent paragraphs; 4. Literature overview...
Responses:
- Figure #1 – Decision tree of Dialysis Treatment Modalities. Included towards the end of “ Chronic Kidney Disease.”
- Figure #2 – Interconnectedness of the effects of relocation and health outcomes. Included towards the end of “ Literature Review.”
- Figure #3 - Illustrates the significance of a multidisciplinary team as it promotes a holistic approach to patient care. Included towards the end of “ Discussion”
- Restructure the paper – While the authors restructured the article as per recommendations, the authors chose to format the paper as follows:
- Introduction (and objective)
- Chronic Kidney Disease
2.1. Stages of Chronic Kidney Disease
2.2. Forms of Treatment
2.2.1. Hemodialysis (HD)
2.2.2. Peritoneal Dialysis (PD)
- Dialysis and Northern First Nation Communities
3.1. Quality of Living
3.2. Geographical Location
3.3. Medical Relocation
3.4. Selection of Dialysis Treatment
- Method
- Literature Review
5.1. Cultural Isolation
5.2. Alienation from family and friends
5.3. Somatic Issues
5.4. Psychosocial Issues
5.5. Loss of Independence and Role Adjustment
- Discussion
- Future Implications
- Conclusions
The authors decided to restructure the article as indicated above as Sections 1-3 provide background information whereas the Method of the paper is typically discussed prior to the Literature Review.
Reviewer: 2
Comments to the Author:
The authors of this manuscript presented the psycho-social and somatic effects of the relocation of indigenous people in Canada, who have chronic kidney disease (CKD).
This article needs to be improved.
- Please correct the title. This article shows the 'Psycho-social and Somatic Effects of Relocation ...'
Abstract
Please edit the summary.
- Please delete the first sentence that suggests this article may be about diabetes mellitus.
- Please improve the purpose of the study. What effects of relocation do you mean ? It is a worsening of kidney tests or an increase in mortality ? (No)
Introduction
- Subsection 1.2 must be completed with CKD diagnostic criteria and a description of CKD stages.
- Subsection 1.3 should be shortened (this is not an article on dialysis methods). Items 1.3.1- 1.3.2 should be summarized in three-four sentences.
- Point 1.4 should be a separate chapter. Break this chapter down into three subsections.
Conclusions
- The conclusions should be more related to the purpose of the study.
Responses:
- Recommendation #1: (Line 2-4) – Title Revision (as recommended to make the title more specific to the review)
- “The Psychosocial and Somatic Effects of Relocation from Remote Canadian First Nation Communities to Urban Centers on Indigenous Peoples with Chronic Kidney Disease (CKD).”
- Additional changes made to the title were with respect to language/terminology (i.e., First Nation Communities instead of reserves).
- Recommendation #2: (Line 9-10) – Removed the first sentence of the abstract as it is suggestive that the review is about diabetes mellitus
- The sentence now reads: “Chronic Kidney Disease, also referred to as End Stage Rental Disease (ESRD), is a prevalent and chronic condition for which treatment is necessary as a means of survival once the affected individuals reach the fifth and final stage of the disease.”
- Recommendation #3: (Line 19-21) – Improve the purpose of the study
- The objective of the study is now more clearly defined: “Consequently, the objective of this paper is to determine the psychosocial and somatic effects of relocation from Canadian First Nation Communities to urban centres on Indigenous Peoples with Chronic Kidney Disease (CKD).”
- Recommendation #4: (Line 55-65) – Include CKD diagnostic criteria and description of CKD stages
- The following information was added to the review: “Additional indicators of CKD (which must be present for approximately three months) include the presence of albuminuria in urine, urine sediment abnormalities, electrolyte imbalances, histologic abnormalities, structural abnormalities, and/or history of kidney transplant” [36].
- “For further information regarding the stages of CKD and the diagnostic criteria, please refer to the KDIGO 2012 clinical practice guideline for the evaluation and management of chronic kidney disease [36] (p. 5).”
- Purpose of change: While the authors did include additional information regarding the diagnostic criteria for CKD, the authors chose to refer readers to the Clinical Practice Guideline as this can more accurately provide medical information to the reader.
- Recommendation #5: (Line 66-129) – Shorten
- We have shortened the section as per recommendations. Please refer to paper for further details.
- Recommendation #6: (Line 130-212) – Move section 1.4 into a separate chapter and break it down into subsections
- Note: Section 1.4 has now been changed to section 3. It has also been renamed as follows: “Dialysis and Northern First Nation Communities”
- The breakdown of this chapter is as follows:
- Dialysis and Northern First Nation Communities
3.1. Quality of Living
3.2. Geographical Location
3.3. Medical Relocation
3.4. Selection of Dialysis Treatment
- Recommendation #7: (452-462) – Revise conclusion so it is more related to the purpose of the study
- We chose to revise the entire conclusion – please refer to the manuscript for details.
Additional Revisions made by the Authors:
- Typographical errors were noted in a section of the paper (See lines 250-358) – changes were made via track changes.
- Authors removed the word “Aboriginal” and “reserves” throughout the article and instead chose to use “Indigenous Peoples” and “First Nation Communities” to reflect cultural sensitivity.

Reviewer 2 Report
The authors of this manuscript presented the psycho-social and somatic effects of the relocation of indigenous people in Canada, who have chronic kidney disease (CKD).
This article needs to be improved.
1.Please correct the title. This article shows the 'Psycho-social and Somatic Effects of Relocation ...'
Abstract
Please edit the summary.
2. Please delete the first sentence that suggests this article may be about diabetes mellitus.
3. Please improve the purpose of the study. What effects of relocation do you mean ? It is a worsening of kidney tests or an increase in mortality ? (No)
Introduction
4. Subsection 1.2 must be completed with CKD diagnostic criteria and a description of CKD stages.
5. Subsection 1.3 should be shortened (this is not an article on dialysis methods). Items 1.3.1- 1.3.2 should be summarized in three-four sentences.
6. Point 1.4 should be a separate chapter. Break this chapter down into three subsections.
Conclusions
7. The conclusions should be more related to the purpose of the study.
Author Response

(The authors gave the same response as above.)
